# Oxidative Stress Evaluation in Dogs Affected with Canine Monocytic Ehrlichiosis

**DOI:** 10.3390/antiox11020328

**Published:** 2022-02-08

**Authors:** Michela Pugliese, Vito Biondi, Giordana Merola, Alessandra Landi, Annamaria Passantino

**Affiliations:** 1Department of Veterinary Sciences, University of Messina, Polo Universitario Annunziata, 98168 Messina, Italy; vito.biondi@unime.it (V.B.); giordana.merola@gmail.com (G.M.); passanna@unime.it (A.P.); 2Veterinary Practitioner, 98168 Messina, Italy; alelandi1612@gmail.com

**Keywords:** *Ehrlichia canis*, reactive oxidative metabolites, antioxidant barrier, ratio between R-reactive oxidative metabolites and antioxidant barrier, thiol groups of plasma compounds

## Abstract

The study aimed to evaluate the concentration of reactive oxidative metabolites (R-OOHs), the antioxidant barrier (OXY), and the ratio between R-OOHs and OXY (OSi) and thiol groups of plasma compounds (SHp) in in canine monocytic ehrlichiosis. Thirty dogs affected with monocytic ehrlichiosis (canine monocytic ehrlichiosis group—CME group) and ten healthy dogs (control group—CTR group) were evaluated. CME was diagnosed by the presence of clinical signs and the detection of anti-*Ehlichia canis* antibodies. Oxidative stress parameters of two groups were compared using the Mann–Whitney test. Significance was set at *p* < 0.05. Spearman rank correlation was performed to analyze oxidative stress, and hematological and biochemical variables in the CME group. All dogs affected with CME showed a wide spectrum of clinical signs such as lethargy, anorexia, fever, weight loss, lymph adenomegaly, splenomegaly, subcutaneous and mucosal petechial and ecchymosis, and vomiting. Anaemia, leukocytosis, thrombocytopenia, hyperglobulinemia, hypoalbuminemia and an increase of blood urea nitrogen and creatinine are also detected. Results showed significantly lower values of SHp in the CME group than in CTR. A statistically significant difference in the number of white blood cells, platelets, and blood urea nitrogen concentration was assayed comparing to the two groups. A negative correlation between SHp and hemoglobin concentration was recorded. These preliminary results may suggest a possible function of oxidative stress in the onset of clinical signs during the course of CME.

## 1. Introduction

Canine monocytic ehrlichiosis (CME) is a tick-borne disease with global distribution, caused by *Ehrlichia canis,* a gram-negative, obligate intracellular bacterium [1,2,3]. 

*Rhipicephalus sanguineus*, commonly called the brown dog tick, is the vector of disease [4,5].

In experimental conditions, after an incubation period of 8–20 days succeeding tick bite, the course of CME can be consecutively differed into acute (2–4 weeks duration), subclinical (months to years) and chronic phases, although the difference among these phases is not straightforward in dogs with naturally-occurring disease [2]. 

Clinical signs such as hyperthermia, anorexia, weight loss, edema, hemorrhage, lymph adenomegaly, splenomegaly, mucosal pallor, uveitis and blindness, mild anemia, thrombocytopenia, and leukopenia may be wide according to the clinical phase of asymptomatic, acute, or chronic infection [6]. In the chronic phase, symptoms of mucosa oral involvement (ulcerative stomatitis and necrotic glossitis) and of the central nervous system (seizures, ataxia, vestibular dysfunction, and cervical pain), posterior limb and/or scrotal edema are also reported. Bleeding diathesis may appear in both the acute and chronic phases of CME, but is more described in the chronic phase, manifesting as cutaneous and mucosal petechiae and ecchymoses, epistaxis, hematuria, melena, and prolonged bleeding from venipuncture sites. In the subclinical stage of CME, clinical manifestations and/or hematological abnormalities may not be detected, or be present in a mild form (e.g., splenomegaly, intermittent fever, thrombocytopenia, anaemia) [2]. The presence of concurrent infection with other vector-borne pathogens or co-morbidities in dogs is considered by clinicians as infrequent or uncharacteristic disease signs. It has been suggested that oxidative stress may play a central function in the onset of clinical signs in the course of CME [7,8], as well as in another vector-borne disease, canine leishmaniasis [9]. Previous studies performed in experimental conditions have demonstrated a status of oxidative stress in dogs infected by *Ehrlichia canis* [10,11]. Changes in markers of antioxidant defense and oxidative damage including increased lipid and protein oxidation are described [8]. Although oxidative stress is involved in the pathogenesis of CEM, the role of this process in the onset of clinical signs related to kidney and liver dysfunction and haematological alterations are not examined. Furthermore, few are studies involving the determination of oxidative stress and the oxidant capacity in dogs naturally infected with CEM [11]. Considering the above-mentioned, the present study aimed to evaluate the concentration of reactive oxidative metabolites (R-OOHs), the antioxidant barrier (OXY), and the ratio between R-OOHs and OXY values (OSi) and thiol groups of plasma compounds (SHp) in dogs affected by CME. The correlation between these variables and haematological and biochemical indicators of hepatic and/or renal functions have been also investigated.

## 2. Materials and Methods

### 2.1. Ethical Statement

The study was conducted according to the standards set by the European Council Directive 2010/63/EU on the protection of animals used for scientific purposes, and the Italian legislation (D.lgs. 26/2014, L. n. 281/1991 and L.R. 15/2000). The Institutional approval from the Ethics Committee of the Department of Veterinary Sciences, University of Messina (Approval number: 49 of 21 March 2021), was also obtained. Dogs were recruited at a shelter belonging to an animal protection organization in Messina (Sicily, South Italy) during the sanitary routine visit. Dog Shelters’ Administration was informed on research objectives and clinical procedures and signed informed consent before samples and data collection, as based on the national guidelines for animal welfare and data protection. 

### 2.2. Animals

Thirty (17 males and 13 females) adult mixed-breed dogs housed in a shelter with acute infection of canine monocytic ehrlichiosis (CME group) were included. The body weight was between 11 and 34 kg (mean 19.3 kg ± 10.5). The mean age was 3.2 ± 2.5 years old. 

Dogs belonging to the CME group were considered affected with ehrlichiosis based on the presence of several clinical signs, such as apathy, appetite loss and intermittent fever [12,13], associated with detection of anti-*Ehlichia canis* antibodies with the indirect immunofluorescence assay test (IFAT). The detection of IgG antibodies was carried out using commercial kits following the manufacturers’ instructions (Fuller Laboratories Fullerton, CA, USA). An antibody titre of 1:50 or greater was considered positive for *E. canis* antigens. Positive and negative controls were added to each slide. Two-fold serial dilutions were set and tested to delimit the serum titre of samples testing positive at screening.

The control group (CTR) was constituted by ten adult mixed breed dogs seronegative for *E. canis* (IFAT), considered healthy on physical examination, and haematological and biochemistry evaluations performed during routine check-ups or vaccinations. 

All dogs were considered negative for *Dirofilaria immitis* antigen, antibodies against *Anaplasma phagocytophilum*, and for antibodies against *Leishmania*, using rapid diagnostic ELISA test kits (SNAP 4Dx®, IDEXX Laboratories, Westbrook, ME, USA).

Furthermore, they had not received any treatment in the last 30 days before the onset of the sampling. These infections were excluded using specific IFAT and/or PCR. 

Hematological and biochemical tests indicating liver and/or kidney injuries were also performed in animals belonging to two groups.

### 2.3. Collection of Blood Samples

Blood samples were collected at the same time (09.00 a.m.) by jugular venepuncture with a vacutainer blood collection system. Blood was divided into two tubes: one without additive and another with tripotassium ethylenediaminetetraacetic acid (K_3_EDTA) (S-Monovette® Sarstedt). After the clotting and centrifugation (ALC 4235 A, Milan, Italy) at 3000× *g*, 20 min at room temperature, the sera obtained was stored at −80 °C until the analysis. Blood smears were also set and stained for microscopic examination.

### 2.4. Samples Analysis 

Oxidative stress parameters (R-OOHs, OXY, and SHp) were measured with an ultraviolet spectrophotometer (Slim SeaC, Florence, Italy) with a “spin traps” technique, based on the identification of coloured products resulting from the reaction between oxygen metabolites and free radicals. 

R-OOHs were evaluated considering the levels of hydroperoxides (R-OOH), produced by peroxidation of lipids, amino acids, proteins, and nucleic acids during the tissue damage, and molecules photometrically revealed after reaction with a properly buffered chromogen. Values directly correlated to the colour amount were displayed in Carratelli units (1 CaRR U = 0.08 mg% hydrogen peroxide).

OXY was assessed by the concentration of residual unreacted radicals detected by the oxidizing effect of a surplus of hypochlorous acid with the plasma inside a water solution. A reduction of values is directly correlated with the modification of the plasma barrier due to oxidation, and thus to the gravity of the damage. If the excess of radicals of hypochlorous acid consequent to considerable oxidation is elevated, the plasma barrier is reduced, and vice versa. SHp was evaluated by the capability of thiol groups to produce a coloured complex when reacted with DTNB (5,5-dithiobis-2-nitrobenzoic acid). Low values are strongly associated with reduced efficiency of the thiol antioxidant barrier. 

The oxidative stress index (OSi) considered as an index of plasma redox status was calculated as the ratio between the values of d-R-OOHs and OXY (×100) [14].

Haematological examination was performed on each K_3_EDTA sample using a haematology analyzer (Procyte Dx, IDEXX Laboratories, Inc., Westbrook, ME, USA). Clinical chemistry profiles on serum aliquots obtained from centrifugation were performed with a chemistry analyzer (Catalyst Dx, IDEXX Laboratories, Inc., Westbrook, ME, USA) to determine blood urea nitrogen (BUN), creatinine (CREA), total proteins (TP), alanine-amino transferase (ALT), aspartate-amino transferase (AST), albumin (ALB) and globulin (GLOB). All samples were analyzed in duplicate by the same operator. 

### 2.5. Statistical Analysis

Statistical analysis was executed with an SPSS for Windows package (Version 17.0, SPSS, Inc., Chicago, IL, USA). Data were expressed as the mean ± standard deviation (SD). The distribution of the data was verified by the Kolmorogov–Smirnov test. An unpaired t-test was performed to assess the differences between CME and CTR groups. Spearman’s rank test was used for the determination of the relationship between the oxidative stress parameters and haematological and biochemical variables. The level of significance was set at *p* = 0.05. 

## 3. Results

All dogs belonging to the CME group presented several clinical signs, including depression (28/30; 91.6%), anorexia (23/30; 69%), fever (25/30; 75%), weight loss (18/30; 54%), lymph adenomegaly (21/30; 63%), splenomegaly (18/30; 54%) subcutaneous and mucosal petechial and ecchymosis (11/30; 36.3%), and vomiting (4/30; 13.3%). 

Alterations in laboratory analytes included anaemia, leucocytosis, thrombocytopenia, hyperglobulinemia, hypoalbuminemia, an increase of BUN, and CREA.

Values for each variable are expressed as mean and SD (Table 1 and Table 2). Dogs of the CME group exhibited statistically lower values of SHp (*p* < 0.001, 63.6 ± 16.8 vs. 267.1 ± 139.8) and statistically higher values of OSi (*p* = 0.002; 76.6 ± 80 vs. 49.2 ± 20.1) compared to the CTR group. No significant differences were detected for R-OOHs and OXY values (Table 2). A statistically significant difference in white blood cells (WBC) (*p* = 0.044), platelets (PLT) (*p* < 0.001), and blood urea nitrogen (BUN) (*p* = 0.045) concentration was assayed, comparing the two groups. (Table 2). 

Spearman’s correlation and regression (R^2^) analysis of data acquired from the CME group presented a negative correlation between R-OOHs and SHp (*p* = 0.002; r = −0.48). A positive correlation was detected between R-OOHs and OXY (*p* = 0.03; r = 0.66). A positive correlation was recorded between SHp and Hgb (*p* = 0.05; r = 0.45), and OSi and R-OOHs (*p* < 0.001; r = 0.81); a negative correlation between OSi and OXY (*p* < 0.001; r = −0.72), OSi and PLT (*p* < 0.01; r = −0.61), and SHp and Hgb (*p* = 0.05; r = −0.45). No correlation was recorded between haematological variables and indices of liver and kidney damages. Data are summarized in Table 3 and Table 4.

## 4. Discussion

The strict intracellular location and lifecycle connection with arthropods are the most common phenotypic characteristic of bacteria of order Rickettsiales (as *Ehrlichia canis*), showing a specific affinity for vascular endothelial cells of small- and medium-sized blood vessels [17,18].

Following the infection, the endothelial cells exhibit structural changes of the endoplasmic reticulum–nuclear envelope induced by oxidative stress, with subsequent accumulation of intracellular peroxides and superoxide radicals [19]. As reported in our results about the CME group, a reduction of cellular thiols associated with high detection of extracellular H_2_O_2_ is described in the course of *Ehrlichia canis* infection [20].

In this study, significant low values of SHp in the CME group were assayed. Thiols are a group of organic sulfur derivatives, characterized by the presence of a sulfhydryl group (-SH) [21]. The thiol redox status of intracellular and extracellular parts plays an important role in the definition of protein structure, control of enzyme activity, and regulation of transcription activity. Thiol antioxidants are involved in different mechanisms. They act as constituents to the thiol/disulfide redox action, metal chelators, radical quenchers, substrates of redox reactions, and specific reducers of protein disulfate connections (thioredoxin) [22]. The conformation and redox status of thiols in a specified compartment is highly mutable, given that it influences the metabolic activity of each compartment and determines the advantageous disposal of specific antioxidants under oxidant stress states. Thiol is considered a novel biomarker of balance of both intracellular and extracellular oxidative processes. Thiol values are inversely proportional to oxidative imbalance [23]. The thiol deficiency is described in various viral and bacterial diseases; conversely, their activity seems to be increased during the course of parasitic diseases sustained by protozoa such as *Leishmania* spp. [9]. Probably, the difference mentioned above is related to different mechanisms involved in the determinism of diseases. It has been reported that oxidant levels increase as thiol oxidation decreases in response to the increasing levels of a total oxidant status [24,25,26,27]. The CME group showed lower values of SHp than the CTR group, suggesting active oxidative stress in dogs affected by ehrlichiosis.

OSi offers a comprehensive evaluation of oxidative stress degree. It is well known that high values of OSi reveal a discrepancy between oxidant and antioxidant structures, with an excess of R-OOHs that alters the structure of the cellular membrane, determining alterations in their fluidity [28]. Indeed, higher values of OSi were observed in dogs affected by the CME than in the CTR group. In addition, a significant correlation was detected between OSi and R-OOHs and between OSi and SHp, while a negative correlation was found between OSi and OXI in the CME group.

Leucocytosis, anaemia, thrombocytopenia, and an increase of urea are common laboratory abnormalities reported during the course of CEM [2,12]. A complete blood count is fundamental in the diagnosis of CME. During the acute stage of disease, moderate to severe thrombocytopenia is a characteristic hematological abnormality. Generally, thrombocytopenia in the acute phase is associated with mild anaemia. During the subclinical phase, mild thrombocytopenia may be present in the absence of other clinical signs. In the chronic phase, thrombocytopenia is usually severe and accompanied by marked platelet counts, and their degree have been suggested as a screening test for CME in endemic regions [12].

Anaemia and thrombocytopenia are generally accepted as principal hematological features consequent to infection of CME [6]. Dogs in the CME group showed significantly lower values of Hgb and PCV, suggesting the presence of anaemia. Anaemia is clinically defined as a reduction of haemoglobin, low haematocrit and/or a low erythrocyte count, and is usually described in dogs in different phases of CME [7,8,13,29]. Various mechanisms for anaemia manifestation have been suggested in dogs with CME. Mild or moderate hypochromic microcytic anaemia is reported in dogs with CME, consequent to a decrease of hemosiderin deposition in bone marrow, typically indicating low serum iron concentrations despite adequate iron stores [30]. Other immunological mechanisms involved consider the destruction of erythrocytes, through the production of antibodies binding on cellular membranes [31]. The negative correlation between the Hgb concentration and SHp could suggest that the onset of anaemia may be related to the oxidative process.

Platelets are implicated in primary hemostasis, aggregating at places of vascular insult to form a hemostatic clot and determining the cessation of bleeding. In platelets, substances reacting with cell surface sulfhydryl units are efficient in blocking numerous functional reactions such as integrin-mediated aggregation, adhesion, and granule secretion [32,33]. Thrombocytopenia is the cause of several clinical signs observed in course of CME, such as skin and subcutaneous hemorrhages of pale visceral organs, ascites, and jaundice. Though different factors are reported as causes of thrombocytopenia in dogs with CME, the main cause lies in a significant increase in levels of serum antiplatelet IgG, subsequent to a reduction in circulating half-life of platelets and acute destruction of platelets [34]. The negative correlation between OSI and platelet counts reported in dogs with CME suggest a potential role of oxidative stress in platelet depletion. The stress oxidative occurs when the oxidant potential dominates antioxidant potential, following an increasing release of free oxygen radicals, a decreasing elimination of free oxygen radicals and insufficiency of antioxidants [35]. The oxidative stress of platelet cell membranes determines a loss of cell membrane elasticity, an increase of fragility, an impairment of selective absorptivity, alterations of receptors, and consequently the life duration of cells shortens. Moreover, protein disulfide isomerase enzyme deriving from platelets does not function properly when thiol is lacking, causing a reduction of platelet number [36].

## 5. Conclusions

The dogs infected by *Ehrlichia canis* probably reacted to oxidative stress, reducing the activity of thiols. We can conclude that there is an increase in the levels of OSI and a decrease of SHp in dogs naturally affected by CME. Results indicate that CME determines oxidative stress, and this disparity between the production and elimination of oxidants in the organism may contribute to the pathophysiology of disease and the onset of hematological findings. Therefore, it is possible to hypothesize that treating the oxidative stress present in dogs affected by CEM with antioxidant supplementation may improve the clinical outcomes of disease.

## Figures and Tables

**Table 1 antioxidants-11-00328-t001:** Concentration of R-OOHs, OXY and SHp in CME and CTR Groups.

Variable	Unit	CME		CTR		*p*-Value
		Mean	SD	Mean	SD	
R-OOHs	(U CARR)	211.9	89.9	247.2	79	Ns
OXY	(μmol HCLO/mL)	454.2	177	419.2	191	Ns
SHp	(μmol/L)	63.6	16.8	267.1	139.8	<0.01
OSi	%	76.6	80	49.2	20.1	0.02

R-OOHs, reactive oxidative metabolites; OXY, antioxidant barrier; SHp, thiol groups of plasma compounds; OSi, oxidative stress index; Ns, not significant.

**Table 2 antioxidants-11-00328-t002:** Haematological and biochemical variables in dogs affected by CME (CME group).

Variable	Unit	CME		CTR		Reference Ranges [15,16]	*p*-Value
		Mean	SD	Mean	SD		
RBC	(10^6^/μL)	6.18	2	5.9	2	5.6–8.7	Ns
Hgb	(ng/mL)	12.9	1.9	15	2	14.7–17.7	0.049
PCV	(%)	38.2	5	43	6	42–53	0.048
WBC	(10^3^/μL)	13.2	4.4	10	1.9	4.6–10.6	0.044
PLT	(10^3^/μL)	127	115	290	99	150–400	<0.01
BUN	(mg/dL)	32.6	12.8	24.4	9.7	5–21	0.045
CREA	(mg/dL)	1.4	0.5	1.2	0.4	0.3–1.2	Ns
AST	(U/L)	43.5	28.4	35.6	23.2	0–40	Ns
ALT	(U/L)	39.4	18.4	34.8	16.6	0–40	Ns
ALB	(g/dL)	3.5	0.62	3.7	0.48	3.0–4.4	Ns
GLOB	(g/dL)	3	0.01	2.7	0.48	1.8–3.9	Ns
TP	(g/dL)	6	0.2	6.4	0.5	6.4–7.9	Ns

RBC, red blood cells; Hgb, haemoglobin; PCV, packed of cells volume; WBC, white blood cells; PLT, platelets; BUN, blood urea nitrogen; CREA, creatinine; AST, aspartate-amino transferase; ALT, alanine-amino transferase; ALB, albumin; GLOB, globulin; Ns, not significant.

**Table 3 antioxidants-11-00328-t003:** Correlation between different oxidative stress variables in the CME Group (Spearman rank order) (* *p* < 0.05; ** *p* < 0.01).

	R-OOHs	OXY	SHp	OSi
R-OOHs	-	−0.27	−0.48 *	0.81 **
OXY	−0.27	-	0.66 **	−0.72 **
SHp	−0.48 *	0.66 **	-	0.66 **
OSi	0.81 **	−0.72 **	0.66 **	-

R-OOHs, reactive oxidative metabolites; OXY, antioxidant barrier; SHp, thiol groups of plasma compounds; OSi, oxidative stress index; * statistically significant.

**Table 4 antioxidants-11-00328-t004:** Correlation between oxidative stress and haematochemical variables in the CME group (Spearman rank order) (* *p* < 0.05).

	RBC	Hgb	PCV	WBC	PLT	ALT	AST	BUN	CREA	ALB	GLOB	TP
R-OOHs	0.26	0.57	0.97	0.73	0.92	0.05	−0.04	0.35	0.33	−0.21	0.01	−0.17
OXY	−0.19	−0.06	−0.02	0.33	0.35	−0.12	0.28	0.13	0.25	−0.11	0.30	−0.39
SHp	−0.31	−0.45 *	−0.34	−0.30	0.02	0.02	0.03	0.08	−0.15	0.03	−0.05	0.34
OSi	0.04	0.04	0.15	−0.2	−0.61 *	0.19	−0.11	0.09	0.06	0.26	0.27	0.12

R-OOHs, reactive oxidative metabolites; OXY, antioxidant barrier; SHp, thiol groups of plasma compounds; OSi, oxidative stress index; RBC, red blood cells; Hgb, haemoglobin; PCV, packed of cells volume; WBC, white blood cells; PLT, platelets; BUN, blood urea nitrogen; CREA, creatinine; AST, aspartate-amino transferase; ALT, alanine-amino transferase; ALB, albumin; GLOB, globulin; * statistically significant.

## Data Availability

Data is contained within the article.

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
