# Peer review of "Oxidative Stress Evaluation in Dogs Affected with Canine Monocytic Ehrlichiosis"

_antioxidants, 2022, doi:10.3390/antiox11020328_

Round 1
Reviewer 1 Report
The manuscript deals with oxidative stress in canine monocytic ehrlichiosis as part of the pathogenesis. As it is, the manuscript cannot be evaluated in full because (a) the methods are not described in sufficient detail, and (b) the language is in larger parts so poor that it is not possible to comprehend what the authors are trying to state.
The manuscript will profit greatly from language revision by a native professional writer and by revision of the contents in the following points:
- Abstract: important data are missing on clinical signs and hematology/biochemistry that should be included. Abbreviations mentioned for the first time should be written out also in the abstract.
- Keyword: unabbreviated keywords are standard
- Introduction: l. 25f: E. canis is tranmsissted mainly by R. sanguineus which is the brown dog tick (the one and only). The introduction lists previous works on the topic of oxidative stress in CME, so why did the authors want to repeeat thisß A hypoathesis or research question explaining what addtional or novel relsuts were expected would help the reader understand why this study was undertaken.
- Methods: informed consent is not soe much about animal welfare but for data protection. The methods used to evaluated infection status are not described in sufficient detail, were these tests inhouse tests or commercial ones? In the first case a detaieled descrition or reference to such a description shoud be included, for commercial tests the manufacturer should be named according to the journal's guidelines. D. immitis antigen tests are usually commercial point of care tests, not PCR tests. Regarding the sample analysis, the same applies. Methods are not described or referenced for in sufifcient detail - nobody could re-run such protocols from the manuscript.
- Results: What is the difference between depression and lethargy (our protocols for clinical exmaination do not follow sich a distinction, so this confused me)? The Tables sghould be adapted to fit the text width and not be separated by page breaks. the menaing of Ns (not singificant) should be included in the headers of the tables where this was used. There is some inconsistency in the naming of paramtrers, in the Methods blood urea nitrogen is abbreviated "BUN", in the results (text and table) as "UREA" or "BUN". This must be unified.
- Discussion: Ehlichia is a separate genus from Rickettsia, the authors probably refer to a higher order, such as Rickesttsiales. The text is in large parts confusing, the authors should seek support from an English correction service/native speaker. Shortening sentences will improve the flow of the text without loss of information. Since hematology asnd blood biochemistry was also carried out, the additional value of the oxidation parameters for clinical evaluation of the diesease should be discussed, espcecially since the first authors previously published a paper on the clinical significance of some of these parameters in canLeish.
- Reg. language: past tense is usually used throughout a scientific manuscript, here the tenses are mixed, this must be revised.
Author Response
Dear Reviewer,
Thank you very much for your time and all your comments. We have revised the references and increased them. The English structure and grammar of the manuscript have been thoroughly reviewed.
We thank you for precise and thoughtful comments and constructive criticism, which has led to a better manuscript.
We revised the manuscript in relation to the suggestions and more detailed answers are given below.
The changes made in the manuscript to address comments are marked up using the
“Track Changes” function.
Abstract: important data are missing on clinical signs and hematology/biochemistry that should be included. Abbreviations mentioned for the first time should be written out also in the abstract.
Clinical signs and hematology/biochemistry alterations have been added. Abbreviations have been written in extenso (Lines 17-25).
Keyword: unabbreviated keywords are standard
Keywords have not been abbreviated (Lines 27-29)
Introduction: l. 25f: E. canis is tranmsissted mainly by R. sanguineus which is the brown dog tick (the one and only). The introduction lists previous works on the topic of oxidative stress in CME, so why did the authors want to repeeat thisß A hypoathesis or research question explaining what addtional or novel relsuts were expected would help the reader understand why this study was undertaken.
The sentence at line 25 has been modified. A hypothesis of research about the assay of oxidative stress parameters and the correlation with heametological and biochemical alterations has been included in the introduction (Lines 57-61).
Methods: informed consent is not soe much about animal welfare but for data protection. The methods used to evaluated infection status are not described in sufficient detail, were these tests inhouse tests or commercial ones? In the first case a detaieled descrition or reference to such a description shoud be included, for commercial tests the manufacturer should be named according to the journal's guidelines. D. immitis antigen tests are usually commercial point of care tests, not PCR tests. Regarding the sample analysis, the same applies. Methods are not described or referenced for in sufifcient detail - nobody could re-run such protocols from the manuscript.
The sentence about informed consent has been integrated. The names of commercial tests used have been added. The methods are detailed (Lines 84-92).
Results: What is the difference between depression and lethargy (our protocols for clinical exmaination do not follow sich a distinction, so this confused me)? The Tables sghould be adapted to fit the text width and not be separated by page breaks. the menaing of Ns (not singificant) should be included in the headers of the tables where this was used. There is some inconsistency in the naming of paramtrers, in the Methods blood urea nitrogen is abbreviated "BUN", in the results (text and table) as "UREA" or "BUN". This must be unified.
Data about lethargy have been deleted (Line 148). Tables have been adapted to the text width. The abbreviation ‘Ns” was explained in the footnotes (Line 162; 168). The results in the text and in the table have been uniformed.
Discussion: Ehlichia is a separate genus from Rickettsia, the authors probably refer to a higher order, such as Rickesttsiales. The text is in large parts confusing, the authors should seek support from an English correction service/native speaker. Shortening sentences will improve the flow of the text without loss of information. Since hematology asnd blood biochemistry was also carried out, the additional value of the oxidation parameters for clinical evaluation of the diesease should be discussed, espcecially since the first authors previously published a paper on the clinical significance of some of these parameters in canLeish.
The discussion has been modified as requested The genus Rickettsia has been corrected with the order Rickesttsiales (Line 186). The text has been revised and corrected. Shorter sentences have been performed. Haematological and biochemistry are discussed, comparing with a paper previously published from same authors on role of oxidative stress in canLeish (Line 210-214).

Reviewer 2 Report
This is an interesting study linking the pathogenicity of CME to oxidative stress in dogs. The results provide some evidence that CME dogs have increased oxidative stress compared to uninfected dogs, and the authors suggest some interesting mechanisms for how oxidative stress may contribute to the pathogenicity based on their data.
I have some minor suggestions to improve the presentation of the article, and recommend that the English is checked to improve language.
line 25 - delete the word 'mainly'
line 32 - update to '...another vector-borne disease, canine leishmaniasis'
line 38 - correct to CME
line 53 - delete 'of'
line 93 - full stop needed between 'vice versa' and 'SHp'
Table 2 - it could be useful to include a table legend explaining the abbreviations used.
lines 136-140 - please check the values are correct here.
It says "A positive correlation was detected between R-OOHs and OXY (P=0.03; r=0.66)." but the r value given in the table is -0.27.
Also "A positive correlation was recorded between SHp and Hgb (P=0.05; r=0.45)" but in the table the r value is -0.45.
line 141 - Table 4 and 5 should be renamed Table 3 and 4.
Table 4 (line 142) - many of the cells could be shaded as there are a lot of repeated values for each combination of measurements.
line 149 - it should be stated that Rickettsia are in the same Order (Rickettsiales) as Ehrlichia, otherwise the start of the Discussion seems somewhat disconnected from the current study.
line 163 - not sure "stall" is the correct word here.
line 169 - 'Thiol values and are inversely'
line 170 - The CME group showed lower values of SHp than the CTR group
line 178 - should be OXY not OXI
line 203 - oxidative stress
line 205 - "...free oxygen radicals and an insufficiency of antioxidants"
Conclusions - what is the significance of this finding? could treating oxidative stress in infected dogs improve their clinical outcomes?
Author Response
Point.1 - This is an interesting study linking the pathogenicity of CME to oxidative stress in dogs. The results provide some evidence that CME dogs have increased oxidative stress compared to uninfected dogs, and the authors suggest some interesting mechanisms for how oxidative stress may contribute to the pathogenicity based on their data.
I have some minor suggestions to improve the presentation of the article, and recommend that the English is checked to improve language.
Response -
Dear Reviewer,
Thank you very much for your time and all your comments. We have revised the references and increased them. The English structure and grammar of the manuscript have been thoroughly reviewed.
We thank you for precise and thoughtful comments and constructive criticism, which has led to a better manuscript.
We revised the manuscript in relation to the suggestions and more detailed answers are given below.
The changes made in the manuscript to address comments are marked up using the
“Track Changes” function.
line 25 - delete the word 'mainly'
The word has been deleted (line 33).
line 32 - update to '...another vector-borne disease, canine leishmaniasis'
The update has been performed (line 51).
line 38 - correct to CME
The acronym has been corrected (line 57).
line 53 - delete 'of'
“Of” has been deleted (line 80).
line 93 - full stop needed between 'vice versa' and 'SHp'.
The full stop has been added (line 126).
Table 2 - it could be useful to include a table legend explaining the abbreviations used.
A legend has been included (lines 161-162).
lines 136-140 - please check the values are correct here.
Values have been corrected (lines 173-177)
It says "A positive correlation was detected between R-OOHs and OXY (P=0.03; r=0.66)." but the r value given in the table is -0.27.
Also "A positive correlation was recorded between SHp and Hgb (P=0.05; r=0.45)" but in the table the r value is -0.45.
Sentences have been corrected (Line 176).
line 141 - Table 4 and 5 should be renamed Table 3 and 4.
Tables have been renamed (Line 177).
Table 4 (line 142) - many of the cells could be shaded as there are a lot of repeated values for each combination of measurements.
The table has been modified.
line 149 - it should be stated that Rickettsia are in the same Order (Rickettsiales) as Ehrlichia, otherwise the start of the Discussion seems somewhat disconnected from the current study.
The sentence has been modified (Line 186).
line 163 - not sure "stall" is the correct word here.
The word has been changed in “compartment” (Line 201).
line 169 - 'Thiol values and are inversely'
The word has been deleted.
line 170 - The CME group showed lower values of SHp than the CTR group.
The sentence has been modified (line 201).
line 178 - should be OXY not OXI (line 219).
The acronym has been corrected.
line 203 - oxidative stress.
The sentence has been modified (line 252).
line 205 - "...free oxygen radicals and an insufficiency of antioxidants".
The sentence has been modified (line 255).
Conclusions - what is the significance of this finding? could treating oxidative stress in infected dogs improve their clinical outcomes?
The appropriate conclusion, relating to the possibility to treat the oxidative stress of dogs affected by canine monocytic ehrlichiosis to improve the clinical signs, is added (267-270).
.
